# In Vivo Anatomical Research by 3D CT Reconstruction Determines Minimum Acromiohumeral, Coracohumeral, and Glenohumeral Distances in the Human Shoulder: Evaluation of Age and Sex Association in a Sample of the Chinese Population

**DOI:** 10.3390/jpm12111804

**Published:** 2022-11-01

**Authors:** Xi Chen, Chang Liu, Tangzhao Liang, Jianhua Ren, Shouwen Su, Ping Li, Shaoshen Zhu, Yanbin Chen, You Peng, Weiping He, Shihai Jiang, Kun Wang

**Affiliations:** 1Department of Joint and Trauma Surgery, Third Affiliated Hospital of Sun Yat-Sen University, Guangzhou 510630, China; 2Emergency Department, Mianxian Hospital in Shaanxi Province, Hanzhong 724200, China; 3Dongguan Integrated Traditional Chinese and Western Medicine Hospital, Dongguan 523820, China; 4Institute of Laboratory Medicine, Clinical Chemistry and Molecular Diagnostics, University Hospital Leipzig, 04103 Leipzig, Germany

**Keywords:** shoulder joint, computed tomography, acromiohumeral distance, coracohumeral distance, glenohumeral distance, three-dimensional reconstruction

## Abstract

Accurate measurement of the minimum distance between bony structures of the humeral head and the acromion or coracoid helps advance a better understanding of the shoulder anatomical features. Our goal was to precisely determine the minimum acromiohumeral distance (AHD), coracohumeral distance (CHD), and glenohumeral distance (GHD) in a sample of the Chinese population as an in vivo anatomical analysis. We retrospectively included 146 patients who underwent supine computed tomography (CT) examination of the shoulder joint. The minimum AHD, CHD, and GHD values were quantitatively measured using three-dimensional (3D) CT reconstruction techniques. The correlation between minimum AHD, CHD, and GHD value and age with different sexes was evaluated using Pearson Correlation Coefficient. The mean value of minimum AHD in males was greater than that in females (male 7.62 ± 0.98 mm versus female 7.27 ± 0.86 mm, *p* = 0.046). The CHD among different sexes differed significantly (male 10.75 ± 2.40 mm versus female 8.76 ± 1.38 mm, *p* < 0.001). However, we found no statistical differences in GHD with different sexes (male 2.00 ± 0.31 mm versus female 1.96 ± 0.36 mm, *p* > 0.05). In terms of age correlation, a negative curve correlation existed between age and AHD among the different sexes (male R^2^ = 0.124, *p* = 0.030, female R^2^ = 0.112, *p* = 0.005). A negative linear correlation was found in CHD among the different sexes (male R^2^ = 0.164, *p* < 0.001, female R^2^ = 0.122, *p* = 0.005). There were no differences between age and minimum GHD in both sexes. The 3D CT reconstruction model can accurately measure the minimum AHD, CHD, and GHD value in vivo and is worthy of further investigation for standard clinical anatomical assessment. Aging may correlate with AHD and CHD narrowing for both sexes.

## 1. Introduction

Shoulder joint stability maintenance requires a certain bone-to-bone distance to avoid bony impingement. It has been found that the variations of acromiohumeral distance (AHD), coracohumeral distance (CHD), and glenohumeral distance (GHD) are closely related to the normal function and biomechanical stability of the shoulder joint [1,2,3,4,5,6]. In particular, decreased AHD or CHD is associated with shoulder impingement syndrome development [6,7,8,9,10,11]. Therefore, clarification of a standardly minimum AHD, CHD, and GHD helps a better understanding of the shoulder’s anatomical features. An anatomic specimen shoulder study would be an appropriate approach to measure the minimum scapula-humeral head distance. However, loss of muscular extensibility may affect the accuracy of measurement values. Advances in technology, including high scanning speed, low radiation dose, and stereoscopic visualization for bone, tissue allow CT scans to evaluate the actual distance between the humeral and surrounding bony structures [12,13,14,15,16]. Interestingly, Yoshida et al. [14] and Werner et al. [17] found that the three-dimensional (3D) measurements of AHD were significantly smaller than that of two-dimensional (2D) measurements. By comparing the reliability of measuring shoulder anatomy, further studies proposed that 3D reconstruction measurements can reduce the risk of errors and provide more accurate results than 2D measurements in determining scapula-humeral head distance [12,13]. 

Biological sex-associated anatomical differences exist in the shoulder anatomical features. Giaroli et al. [9] and Yoshida et al. [14] found that the scapula-humeral head distance in females was smaller than that in males. In addition, pioneering research has reported that the average coracohumeral interval for males was 3 mm greater than that for females [9]. Of note, males with shoulder instability were found to have a greater glenohumeral mismatch ratio compared with healthy males. In contrast, female patients with shoulder instability showed no differences in bony glenohumeral morphology compared with controls [18]. To date, the association between sex-based variations of the shoulder anatomic interval and ages has not been extensively studied [19,20,21,22]. 

There are significant skeletal anatomical differences present in different geographical populations, therefore, the humeral head-related bone-to-bone distance could vary widely across geographic groups. Zhang et al. [23] measured the geometric parameters of the shoulder joint in 80 healthy Chinese subjects using 3D reconstruction. It was found that the average radius of humeral head curvature, joint surface diameter, and maximum upper and lower glenoid length in the Chinese population was significantly different from those in the Western population. The same conclusions were obtained by Dipit Sahu et al. [24] and Noboru Matsumura et al. [25], in which the geometric parameters of the shoulder joint were measured in 50 healthy Indians and 160 healthy Japanese subjects, respectively. Additionally, shoulder impingement syndrome was observed in 8.8% of the Finnish population [26], while there were about 18% observed in the Chinese population [27]. In addition, scapula-humeral head distances in Americans, Germans, Japanese, Turkish, and Swedish have been reported previously [14,20,28,29,30,31,32]. To the best of our knowledge, little research has reported the precise 3D measurements of minimum AHD, CHD, and GHD for the Chinese population.

We therefore retrospectively measured the minimum AHD, CHD, and GHD of the healthy shoulder joint in the supine position using our clinical data. By using 3D reconstruction techniques, the minimum AHD, CHD, and GHD in a sample of Chinese healthy shoulders among groups of different sexes would be precisely assessed. Of note, degeneration with age was thought to be the main cause of changes in the scapula-humeral head interval, we, therefore, hypothesized that a negative correlation may exist between age and AHD, CHD, or GHD with a different sex. This retrospective in vivo anatomical study would complement the knowledge of the shoulder anatomical values in a sample of the Chinese population.

## 2. Materials and Methods

### 2.1. Compliance with Ethical Standards

All procedures performed in studies involving human participants were in accordance with the ethical standards of the Third Affiliated Hospital of Sun Yat-sen University ethics committee and with the 1975 Helsinki Declaration and its later amendments or comparable ethical standards. The protocol was previously reviewed and approved by the Third Affiliated Hospital of Sun Yat-sen University Ethics and Research Committees ([2022] 02-013-01).

### 2.2. Patient Data Study

Between January 2011 and December 2021, a total of 1713 patients underwent CT examinations of the shoulder joint. During the study period, all patients had standard shoulder radiographs and supine shoulder CT images taken during their preoperative examination. All images were retrospectively reviewed by two senior orthopedic surgeons (TZL and JHR) and findings were recorded independently. Discrepancies in interpretation were resolved by consensus. 

The inclusion criteria for this study were patients with normal standard shoulder radiographs and supine shoulder CT images. Exclusion criteria were patients with shoulder fractures, rotator cuff tears, shoulder instability, shoulder deformities, shoulder osteoarthritis, upper limb bone tumors, scoliosis deformities, and immunologically related disorders or those aged less than 20 years. In this research, 1399 were excluded because of the history of shoulder fractures, rotator cuff tears, shoulder instability, shoulder deformities, shoulder osteoarthritis, upper limb bone tumors, scoliosis deformities, immunology-related diseases, and poor-quality imaging. Additionally,18 patients were excluded because they were younger than 20 years of age [33]. In addition, previous studies indicated that the distance between the scapula and humeral head would change based on the abduction angle of the upper limb [28,34]. Therefore, 150 patients with upper limb abduction angle (the angle between L1 and L2) and humeral rotation angle (the angle between L4 and L5) greater than or equal to 10° were excluded. (Figure 1) [23,35]. A total of 146 patients were eventually included in the study and these cases were subsequently divided into 12 age groups based on sex and age (Figure 2, Table 1). Both left and right shoulder data were included in the study. Of the 146 patients, left/right shoulder data were collected in 73/73 cases, respectively. Medical records were retrospectively reviewed to collect demographic data including age, sex, and left or right shoulder of the shot at the time of examination.

### 2.3. Evaluation Metrics

Imaging of the shoulders was acquired for each patient using a 64-row CT scanner (Aquilion; Toshiba Medical Systems Corporation, Otawara, Japan). The CT data were accumulated in Digital Imaging and Communication in Medicine (DICOM) data format. 

The 3D bone surface models of the acromion, coracoid process, glenoid cavity, and proximal humerus were extracted from DICOM data using Mimics software (version 20.0; Materialise, Leuven, Belgium) and measured minimum AHD, CHD, and GHD using Meshlab software (version 2020.07; ISTI, Pisa, Italy) [14,23,36]. One-mm-thick axial images were reconstructed using mimics, and the images were segmented and processed by the CT Bone Segmentation program (Figure 3a), followed by the Calculate Part program to reconstruct a 3D surface model of the scapula and proximal humerus (Figure 3b). Finally, we segmented the acromion, coracoid process, and glenoid cavity using the Cut with Curve program (Figure 3c). The bone surface model was generated with the smoothing level setting of two, and the proximal humerus, acromion, coracoid process, and glenoid cavity surface models were exported as Polygon File Format (PLY) data. The minimum distance between the bone surface model of the proximal humerus and the bone surface models of the acromion, coronoid process, and glenoid cavity was automatically calculated as the Hausdorff distance using Meshlab software. This minimum distance indicates that any point in the acromion, coracoid process, and glenoid cavity surface model can be reached at any point in the humeral head surface model by advancing at least the distance [14]. The minimum vertex quality values for the proximal humerus and acromion are recorded as AHD, while the values for the proximal humerus and coronoid process were recorded as CHD and the values for the proximal humerus and the glenoid cavity were recorded as GHD (Figure 4).

### 2.4. Data Process and Statistical Analysis

SPSS Statistics 22.0.0.0 software (IBM Corp., Armonk, NY, USA) was used for the statistical analyses. The data for AHD, CHD, and GHD for males and females showed a normal distribution using the Shapiro–Wilk test (*p* > 0.05). The means and SDs of the geometric parameters were calculated and Bar graphs were plotted to show the distributions of the parameters. Independent-samples t-tests were used to assess whether there was a difference between AHD, CHD, and GHD when the humeral head was internally and externally rotated. The same method was also used to compare differences related to sex, but differences in AHD, CHD, and GHD were evaluated separately for males and females at different ages using the Mann–Whitney U test. Age-related differences were compared using a One-Way ANOVA, if the result was significant, post hoc range tests and pairwise multiple comparisons were applied and the associated line graphs were plotted. Pearson correlation coefficients and curve estimation were used to assess the correlation between the values of AHD, CHD, and GHD with the age of the participants. The results of this study were compared with those of Eastern and Western population cohorts derived by combining several series from the published literature, and the sample sizes, means, and SDs of the individual studies are presented in the tables [14,20,28,29,30,31]. Significance was set as *p* < 0.05. 

## 3. Results

A total of 146 cases with a mean age of 43.97 ± 16.25 years and an equal number of left and right shoulders were included in this study. There were 84 males with a mean age of 41.08 ± 16.42 years and 62 females with a mean age of 47.89 ± 15.28 years. The mean humeral abduction angle was 4.75 ± 3.32° (4.49 ± 3.30° in men and 5.09 ± 3.34° in women). Internal rotation of the humeral head was noted in 71 cases with a mean rotation angle of 5.96 ± 2.87° (6.15 ± 2.89° in men and 5.55 ± 2.83° in women). External rotation of the humeral head was observed in 75 cases with a mean rotation angle of 6.03 ± 2.89° (5.73 ± 2.91° in males and 6.30 ± 2.88° in females) (Table 2). Of the 146 patients, the largest number of patients was in the 30–39 age group (25.3%), and the smallest number was in the >69 age group (7.5%), with approximately equal numbers in the remaining four groups (Table 1). There was no statistical difference in the measurements of AHD, CHD, and GHD at rotations of the humeral head of fewer than 10 degrees (Appendix A), indicating that the data included in this study had a high degree of reliability.

### 3.1. The Minimum Acromiohumeral Distance

The mean value of minimum AHD in males was greater than that in females (male 7.62 ± 0.98 mm versus female 7.27 ± 0.86 mm, *p* = 0.046) (Table 2 and Table 3). The distribution of AHD in males ranged from 6.78 ± 0.49 mm to 7.98 ± 0.81 mm. The distribution of AHD in females ranged from 6.32 ± 0.82 mm to 7.62 ± 0.95 mm. Interestingly, we found a statistical difference in AHD values between males and females only in the age group 30–39 years (*p* = 0.011; Table 3).

In the male groups, AHD values in the 30–39 and 40–49 age groups were significantly greater than in the 60–69 or >69 age groups (*p* < 0.05). In the female groups, AHD values in the 20–29, the 40–49, and the 50–59 years age groups were found significantly greater than that in the >69 years age group (*p* < 0.05) (Figure 5). In terms of age correlation, a negative curve correlation existed between age and AHD both in males (R^2^ = 0.124, AHD = 6.901 + 0.052 × Age − 0.001 × Age^2^, *p* = 0.030) and in females (R^2^ = 0.112, AHD = 6.111 + 0.066 × Age − 0.001 × Age^2^, *p* = 0.005), respectively (Figure 6).

### 3.2. The Minimum Coracohumeral Distance

The mean value of minimum CHD in males was greater than that in females (male 10.75 ± 2.40 mm versus female 8.76 ± 1.38 mm, *p* < 0.0001) (Table 2 and Table 3). The distribution of CHD in males ranged from 8.23 ± 1.27 mm to 11.66 ± 2.38 mm. The distribution of AHD in females ranged from 7.80 ± 0.52 mm to 9.46 ± 1.67 mm. Of importance, we found a statistical difference in CHD values between males and females in the 40–49 (*p* = 0.001) and 50–59 (*p* = 0.047) years old age groups (Table 3).

In the male groups, CHD values in groups before 60 years were found significantly higher than the age group after 60 years (*p* < 0.05). In the female groups, CHD values in group 30–39 years were significantly higher than the age group 50–59 years, 60–69 years, and >69 years, respectively (*p* < 0.05) (Figure 5). Furthermore, a negative linear correlation existed between age and AHD both in males (R^2^ = 0.164, CHD = 13.180 − 0.059 × Age, *p* < 0.001) and in females (R^2^ = 0.122, CHD = 10.275 − 0.032 × Age, *p* = 0.005), respectively (Figure 6). These results suggested that there is a decreasing trend in CHD values with increasing age in both sexes.

### 3.3. The Minimum Glenohumeral Distances

We found that the mean GHD is 2.00 ± 0.31 mm in men and 1.96 ± 0.36 mm in women, respectively, showing no statistical difference (Table 2 and Table 3). In addition, there were no statistically significant differences between age groups of both sexes. Age correlation was not found between GHD values (Figure 5). We, therefore, assumed that age and sex may not be correlated with the variety of the GHD.

## 4. Discussion

In the current study, we accurately measured the minimum AHD, CHD, and GHD values in vivo by using a 3D CT reconstruction model. The primary findings of this study were negative correlation existed between age and AHD, CHD with both sexes, although changes in GHD did not statistically differ by age or sex.

In this study, regression analysis showed that increasing patient age was associated with the narrowing of AHD and CHD, while no significant correlation was seen with GHD. This age-related difference is most evidenced when comparing the shoulders parameters of the young with those of the elderly. Previous results indicated that the shape of the acromion does not change with age, however, the proliferation of bone spurs and aging of the ligaments at the acromion and coracoid process can lead to the narrowing of the subacromial and subcoracoid gaps with age [6,21,22,37,38,39,40]. Ogawa et al. [22] found that bone spurs began to appear in the 20- to 29-year age group and that the overall incidence and size of bone spurs increased with age, with medium and large bone spurs occurring in the 40- to 49-year age group and older subjects. These results coincide with our above findings, which suggested that starting with the 40–49 age group, where AHD and CHD become progressively narrower with increasing age. In addition, we believe that degenerative changes in the ligaments are also important in causing the narrowing of AHD and CHD. In 40 fresh shoulder specimens, Fremerey et al. [38] found a marked effect of aging on the failure load and stiffness of the ligaments, along with both failure load and stiffness being decreased in the older shoulders. Degenerative changes in the coracobrachialis ligaments, therefore, cause an uncontrolled superior displacement of the humeral head and a narrowing of the subacromial space [6,39]. In the present study, no significant correlation was found between the change in GHD values and age, we believe that the size of GHD reflects, to some extent, the thickness of the cartilage in the glenohumeral joint, which is consistent with the findings of Petersson et al. [31], indicating that age is not a major influence on GHD.

Although biological sex-associated anatomical differences exist in the shoulder anatomical features, we did not find a correlation between GHD variance and sex differences. Of note, statistical differences in sex were found only in the 30–39 age group for AHD and in the 40–49 and 50–59 age groups for CHD. In the current study, the mean values for AHD and CHD were significantly greater in men than in women. This is consistent with previous reports that sex differences should be taken into consideration in orthopedic studies [19,41]. In addition, Giaroli et al. [9] found a mean 3 mm smaller CHD in women than in men, whereas in our study only a 2 mm difference was found. We believe this data difference may be related to different rotation angles applied in the humerus possesses at the time of measurement and the differences in the skeletal anatomy of the shoulder that existed in the sample population [23,24,25,28]. In comparison to AHD, a 0.4 mm smaller value found in females compared to males in this study was not significantly different from the finding of Yoshida et al. [14].

Previous studies have shown that there are significant skeletal anatomical differences between populations from different geographical regions [23,24,25]. Compared with previously reported AHD in Germans [29] and Japanese [14], there were significant differences in the measurement of AHD in the present study (Table 4). We believe that this discrepancy in results is not only related to differences in skeletal anatomy but also closely related to different measurement methods. Previous researchers have studied the distance between the scapula and humerus in Western populations using direct measurements in a 2D plane using radiographs and CT scans [20,28,29,30,31]. Nevertheless, according to previous findings, conventional 2D analysis cannot detect the actual nearest point of the scapula and humerus because they have a curved structure on their surface, but this can be observed with 3D imaging [14]. In this study, AHD, CHD, and GHD are the minimum distances between the 3D surfaces of the scapula and humeral head that were assessed by automatic calculation with higher reliability. The CHD values in the present study were similar to those reported by Elif Aktas et al. [32] for Turkish and Richards et al. [20] for Americans. We assume that the coracoid process has a smaller accessible area than the acromion, which allows for less selectivity when measuring CHD in a two-dimensional plane, resulting in stable measurements. Additionally, the CHD measurements in this study were different from those reported by Friedman et al. [30] in the United States. Compared results make us realize that different upper limb placements have an important role in influencing the length of the scapula-humeral head distance. This is in agreement with the findings reported by Evan et al. [28] and Harput et al. [34]. 

From a clinical aspect, variations in the subacromial and subcoracoid space can have an impact on the impingement of the rotator cuff tendons and bursa, which were first described by Neer [3] and Gerber [4]. Researchers noted that patients experienced relief of painful symptoms in the shoulder after decompression surgery was performed to increase the subacromial and subcoracoid spaces [42,43,44]. Narrowing of the space between the rostro-humeral arch and the humeral head can lead to secondary impingement of the shoulder joint, causing reactive, and degenerative bone changes. Such superimposed trauma can lead to rotator cuff rupture, damage to the glenoid labrum, and destruction of articular cartilage, resulting in narrowing of the scapula-humeral head distance and normal functional and biomechanical alterations of the shoulder joint [5,15,21,37]. Burkhart et al. [45] found that the recurrence rate of shoulder instability in patients with or without significant shoulder bone defects was 67% and 4%, respectively, indicating variations in bony structure and bone-to-bone distance could have significant effects on shoulder stability [15]. More importantly, precisely determining the minimum scapula-humeral head distance also has an important effect on designing the anatomic shoulder prosthesis and matching the prosthesis type decision based on the native bony anatomy [46,47].

There are some limitations to this study. Firstly, we report measurement values in 146 patients, which are not sufficient for generalization to the general population, but a multi-center study is being designed in the near future. Secondly, we did not conduct an inter-investigator reliability analysis, which may affect the reproducibility of the results. However, the study was conducted in its entirety by two orthopedic researchers and the measurement of values was entirely automated by the software. Finally, controversy still exists regarding the advantages and disadvantages of different methods of measuring the scapula-humeral head distance, and the prone position also significantly reduces inter-articular distance. Although the standard shoulder joint spacing measurement is variable, we here try a possible accurate method to measure the most realistic joint spacing to provide a theoretical basis for helping the future diagnosis of clinically shoulder joint-relevant diseases.

## 5. Conclusions

The 3D CT reconstruction model can accurately measure the minimum AHD, CHD, and GHD values in vivo and is worthy of further investigation for standard anatomical assessment. Aging may correlate with AHD and CHD narrowing in both sexes in a sample of the Chinese population.

## Figures and Tables

**Figure 1 jpm-12-01804-f001:**
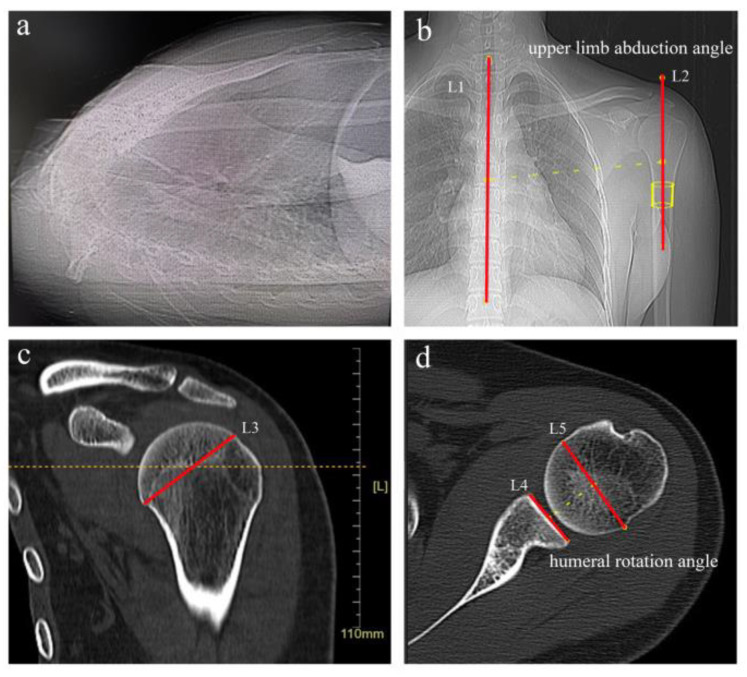
Definition of shoulder measurement axes and angles. (**a**) All subjects were supine in the CT scan tunnel. (**b**) We define the angle between the axis of the proximal humerus (L2) and the line of the spinous process of the spine (L1) as the angle of abduction of the upper limb. (**c**) The level of the center of the humeral head is defined as the level of the median image slice between the most cranial and the most caudal (L3) slice with an identifiable anatomic neck of the proximal humerus. (**d**) We measured the rotation angle of the humerus at the level of the center of the humeral head-the angle between the axis of the humeral head (L5) and the line of bone between the anterior and posterior edges of the glenoid cavity (L4). Patients’ CT scans with upper limb abduction angle (the angle between L1 and L2) and humeral rotation angle (the angle between L4 and L5) greater than or equal to 10° were excluded because of the association between AHD alteration and upper limb abduction angle.

**Figure 2 jpm-12-01804-f002:**
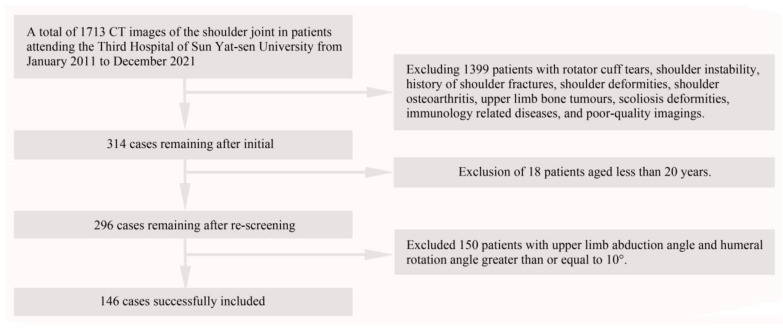
Flowchart diagram of this study. A total of 1713 patients underwent standard CT examinations of the shoulder joint and a total of 146 patients were eventually included in the study.

**Figure 3 jpm-12-01804-f003:**
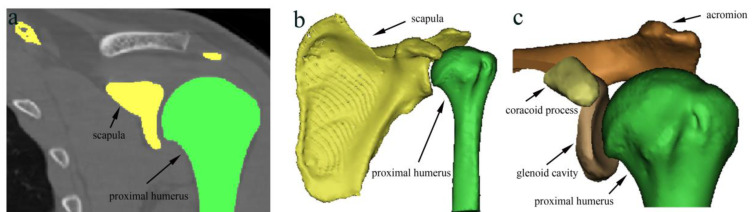
Representative 3D surface models reconstruction and segmentation of CT images of the proximal humerus and scapula. (**a**) Preparation of the proximal humerus and scapula for reconstruction on CT images. (**b**) Bone surface model of the scapula and proximal humerus. (**c**) The 3D surface models of the proximal humerus, acromion, coracoid process, and glenoid cavity. (Not-to-scale).

**Figure 4 jpm-12-01804-f004:**
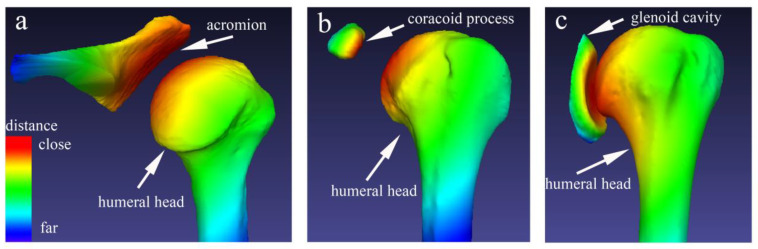
Representative determination of the minimum distance in the 3D surface models. The red areas indicate where (**a**) the acromion, (**b**) the coracoid process, and (**c**) the glenoid with the minimum distance from the humeral head. The sequential color change process from red to yellow, green, and blue represents a gradual increase in the distance between the two surface models. The minimum distance indicates that any point in the acromion, coracoid process, and glenoid cavity surface model can be reached at any point in the humeral head surface model which was automatically calculated. (Not-to-scale).

**Figure 5 jpm-12-01804-f005:**
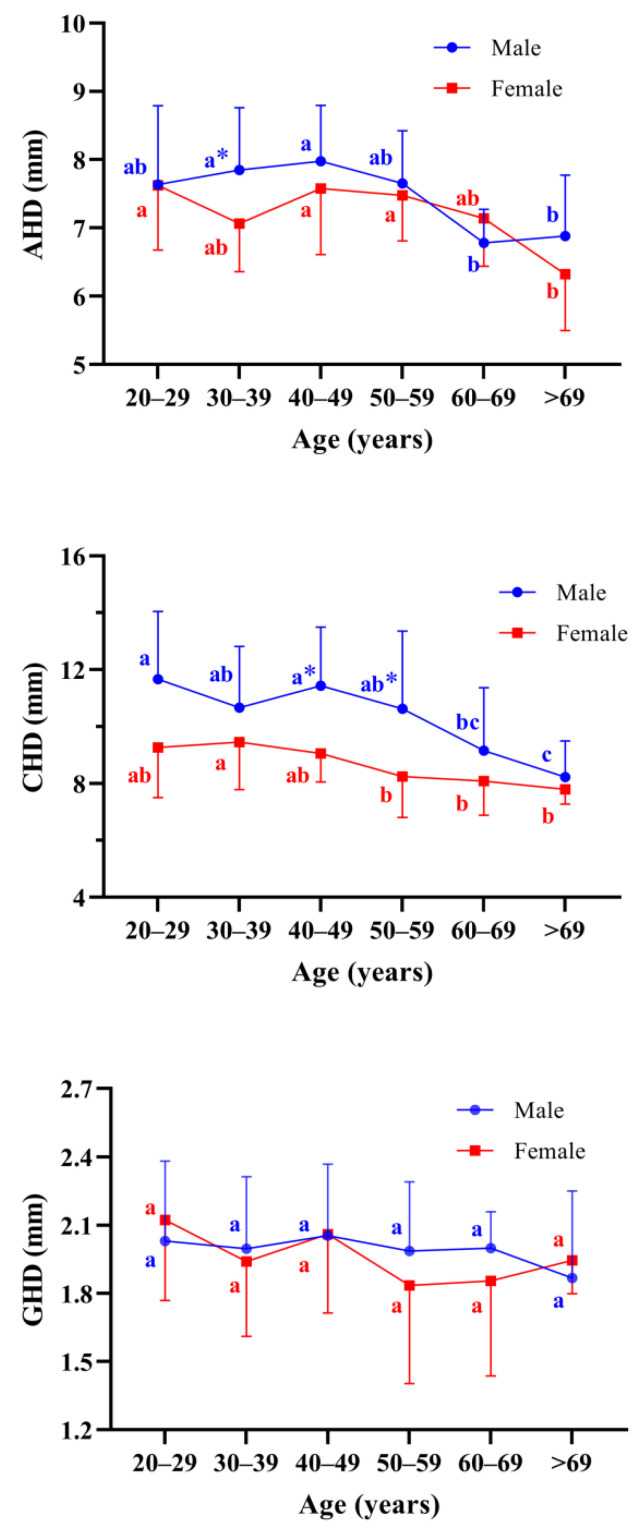
Statistical relationships among ages in both sex and statistical comparation between sexes for AHD, CHD, and GHD. a, b, and c indicate the letter marking method for the comparison between each age group within the group. * Indicates a significant difference in the comparison of the receiving female group *p* < 0.05. AHD acromiohumeral distance, CHD coracohumeral interval, GHD glenohumeral distance.

**Figure 6 jpm-12-01804-f006:**
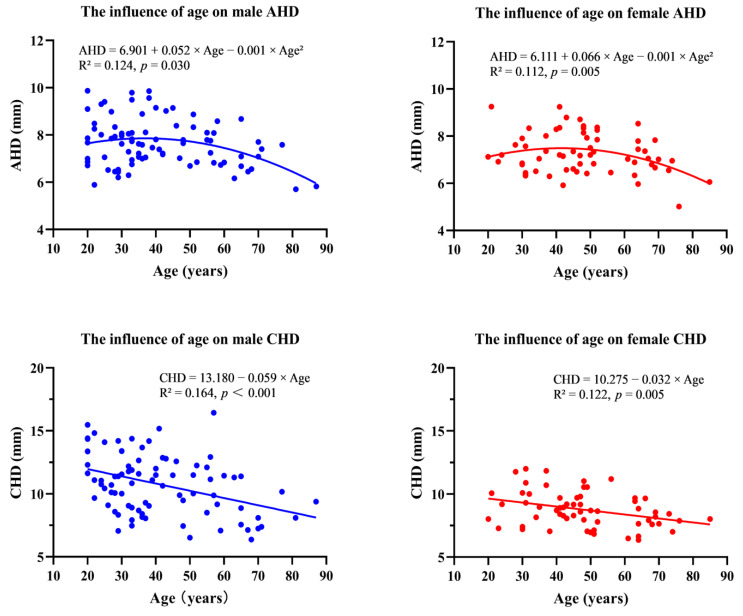
Curve correlation and Linear regression analyses of AHD and CHD. Curve correlation analyses representing the relationships between the CHD and ages in both sexes. Linear regression analyses representing the relationships between the CHD and ages in both sexes. AHD acromiohumeral distance, CHD coracohumeral interval.

**Table 1 jpm-12-01804-t001:** Numbers of observed joints grouped by age and sex.

Sex	Age (Years)						Total
	20–29	30–39	40–49	50–59	60–69	>69	
Male	24	24	11	12	7	6	84
Female	5	13	19	8	12	5	62
Total	29	37	30	20	19	11	146

**Table 2 jpm-12-01804-t002:** Anticipant characteristics.

Characteristic	All	Male	Female	*p* Value (Sex Difference)
Age (years)	43.97 ± 16.25 (n = 146)	41.08 ± 16.42 (n = 84)	47.89 ± 15.28 (n = 62)	0.012
Outreach angles (°)	4.75 ± 3.32 (n = 146)	4.49 ± 3.30 (n = 84)	5.09 ± 3.34 (n = 62)	0.288
Internal rotation angles (°)	5.96 ± 2.87 (n = 71)	6.15 ± 2.89 (n = 48)	5.55 ± 2.83 (n = 23)	0.409
External rotation angles (°)	6.03 ± 2.89 (n = 75)	5.73 ± 2.91 (n = 36)	6.30 ± 2.88 (n = 39)	0.392
Left shoulder (cases)	73	40	33	
Right shoulder (cases)	73	44	29	
AHD (mm)	7.47 ± 0.94 (n = 146)	7.62 ± 0.98 (n = 84)	7.27 ± 0.86 (n = 62)	0.046
CHD (mm)	9.90 ± 2.25 (n = 146)	10.75 ± 2.40 (n = 84)	8.76 ± 1.38 (n = 62)	<0.001
GHD (mm)	1.98 ± 0.33 (n = 146)	2.00 ± 0.31 (n = 84)	1.96 ± 0.36 (n = 62)	0.598

The data are presented as mean ± standard deviation, the number of cases is enclosed in parentheses. AHD acromiohumeral distance, CHD coracohumeral distance, GHD glenohumeral distance.

**Table 3 jpm-12-01804-t003:** AHD, CHD, and GHD statistical analysis between sexes in age groups.

Variable	Age (Years)						*p* Value
	20–29	30–39	40–49	50–59	60–69	>69	
AHD (mm)							
Male	7.63 ± 1.15 (5.89 – 9.87)	7.85 ± 0.91 (6.30 – 9.79)	7.98 ± 0.81 (7.02 – 9.15)	7.65 ± 0.77 (6.69 – 8.87)	6.78 ± 0.49 (6.16 – 7.68)	6.88 ± 0.89 (5.70 – 7.70)	0.039
Female	7.62 ± 0.95 (6.92 – 9.25)	7.06 ± 0.71 (6.30 – 8.34)	7.58 ± 0.96 (5.92 – 9.24)	7.48 ± 0.67 (6.46 – 8.37)	7.14 ± 0.70 (5.97 – 8.53)	6.32 ± 0.82 (5.02 – 7.02)	0.048
*p* Value	0.889	0.011	0.250	0.734	0.261	0.247	
CHD (mm)							
Male	11.66 ± 2.38 (7.06 – 15.47)	10.67 ± 2.14 (7.47 – 14.37)	11.44 ± 2.06 (7.44 – 15.17)	10.63 ± 2.73 (6.52 – 16.43)	9.15 ± 2.21 (6.37 – 11.44)	8.23 ± 1.27 (7.10 – 10.17)	0.011
Female	9.26 ± 1.76 (7.28 – 11.77)	9.46 ± 1.67 (7.05 – 12.01)	9.06 ± 1.00 (7.05 – 11.03)	8.24 ± 1.44 (6.83 – 11.20)	8.08 ± 1.20 (6.36 – 9.68)	7.80 ± 0.52 (7.01 – 8.43)	0.039
*p* Value	0.051	0.124	0.001	0.047	0.432	0.792	
GHD (mm)							
Male	2.03 ± 0.35 (1.47 – 2.75)	2.00 ± 0.32 (1.47 – 2.75)	2.05 ± 0.31 (1.66 – 2.65)	1.99 ± 0.30 (1.52 – 2.67)	2.00 ± 0.16 (1.74 – 2.17)	1.87 ± 0.38 (1.41 – 2.40)	0.903
Female	2.12 ± 0.35 (1.65 – 2.57)	1.94 ± 0.33 (1.41 – 2.43)	2.06 ± 0.35 (1.44 – 2.63)	1.84 ± 0.43 (1.28 – 2.56)	1.86 ± 0.42 (1.24 – 2.42)	1.94 ± 0.15 (1.77 – 2.09)	0.489
*p* Value	0.518	0.649	0.966	0.384	0.773	0.792	

Data are shown as the mean + standard deviation, with the range in parentheses. AHD acromiohumeral distance, CHD coracohumeral distance, GHD glenohumeral distance.

**Table 4 jpm-12-01804-t004:** Comparison of current reported AHD, CHD, and GHD with previous research.

Parameter/Population		Previous Studies			Current Study			*p* Value †
		Mean ± SD	Sample Size	Imaging Method	Mean ± SD	Sample Size	Imaging Method	
AHD (mm)								
Chinese		Not Found			7.47 ± 0.94	146	CT	
Japanese		8.1 ± 1.2	166 [14]	Radio-graphy				<0.001
Japanese		6.6 ± 1.2	166 [14]	CT				<0.001
German		10.4 ± 2.4	234 [29]	Radio-graphy				<0.001
German		9.2 ± 1.8	234 [29]	MRI				<0.001
American	††	11.1 ± 1.4	8 [28]	Radio-graphy				<0.001
CHD (mm)								
Chinese		Not Found			9.90 ± 2.25	146	CT	
Turkish		9.35 ± 2.52	58 [32]	MRI				0.151
American		10.00 ± 1.33	35 [20]	MRI				0.733
American	††	11.00 ± 0.43	50 [30]	MRI				<0.001
GHD (mm)								
Chinese		Not Found			1.98 ± 0.33	146	CT	
Swedish	††	4.68 ± 0.35	175 [31]	Radio-graphy				<0.001

† For the difference between the value in the current study and the one in the previous study or studies (of a Chinese, Eastern, or Western population depending on the row on which the *p* value appears). †† Abduction or rotation angle greater than or equal to 10°. AHD acromiohumeral distance, CHD coracohumeral distance, GHD glenohumeral distance.

## Data Availability

Data are available within the paper and upon reasonable request to the corresponding authors.

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
