# Peer review of "In Vivo Anatomical Research by 3D CT Reconstruction Determines Minimum Acromiohumeral, Coracohumeral, and Glenohumeral Distances in the Human Shoulder: Evaluation of Age and Sex Association in a Sample of the Chinese Population"

_jpm, 2022, doi:10.3390/jpm12111804_

Round 1

Reviewer 1 Report

The article is devoted to the assessment of variation in the acromio-brachial distance (AHD), coraco-brachial distance (CHD) and humeroscapular distance (GHD) in an uneven-aged population of people in China. From a biomechanical point of view, this is important for assessing the movement in the human shoulder joint, as the main one in performing basic "home" manipulations, or, for example, in sports - when greater freedom in arm movement is needed (for example, swimming, tennis) and from a clinical point of view, since functional instability of the shoulder joint can lead to the development of age-related undesirable changes (for example, pain or limitation of mobility), as well as in joint prosthetics.

However, the purpose of the research it is not clear from the Introduction. In my opinion, in the introduction, attention should be paid to the anthropological or anatomical features of the structure of the shoulder joint in the Chinese population, and comparison with the European one. The statistics of the prevalence of joint instability or impingement syndrome of the shoulder joint in the same population should be shown. This can justify the relevance of the study (e.g. 10.1302/2058–5241.5.200049, 10.1016/j.jseint.2020.06.008, and 27th reference).

There is no structure in the Introduction. There are repetition of existing methods of measurements and etc (e.g. lines 47 - 60 on the use of 2D or 3D reconstructions, repeated again in lines 63-70). It should be revised and improved. There are errors in Lines 57-60: os scaphoideum is a part of wrist, not shoulder.

Methods:

Methods should be revised. Figures captions should be expanded: caption should contain total information about Figure. The Figures should be improved to. E.g., in Figure 1 there is no description of the 4th joint section. Lines are too thin. Mentioned angles are not highlighted in the Figures. Moreover, you wrote “Finally, 150 patients with an abduction angle of the upper limb greater than 102 or equal to 10° and an internal and external rotation angle of the humerus greater than or 103 equal to 10° were excluded (Figure 1) [27-29].” But, it is not clear how the Figure can show the information about excluded patients. And what for references [27-29]?

In Figure 3 bones should be named.

In Figure 4 all denotation from text should be shown. In methods there is no description of distance measurement/calculation. The Figure 4 caption should be revised.

Methods should be expanded; there is no information about left and right shoulder analysis. The corresponding information appears only in the Results.

Results:

Results are very detailed. But, then they are duplicated several times in table and figures. Tables and figures duplicate each other. Results should be revised. E.g., Figure 5 and Table 1 (Table 1 is more informative), Figure 6 and Table 3. In Figure 6 the 1st and the 2nd parts are the same, only diagram type is different.

According to presented P value presented gender differences are not statistically significant.

In the discussion, the authors relate interarticular distances to the development of pain, joint instability, and impingement. In my opinion, since the article is more devoted to the anthropological features of ethnic genesis, it can start with this topic. In addition, in the presented manuscript, a healthy population (relative to the shoulder joint) is discussed; therefore, in discussions about possible consequences, secondary forms of the same (e.g., impingement) can be analyzed.

Since the authors did not have their own results of X-ray analysis, and calculations were not made using other methods, the obtained results do not looks like reliable. The authors rightly argue that the geometry of the shoulder joint, the change in the position of the head of the humerus are important in determining the interarticular distance. The prone position also reduces interarticular distance. Perhaps the results obtained can be more important in prosthetics of the shoulder joint (e.g., look articles: 10.2106/JBJS.15.01232, 10.1016/j.jseint.2020.06.008).

Conclusion:

According to the communication with orthopedic the authors leads to the idea that the determination of interarticular distances is important for anatomy. Opposite, for the clinic, functional tests designed to accurately determine the genesis of pain and the formation of impingement, or functional instability of the joint, are of greater importance. Moreover, there are no norms for interarticular distances, that is, this indicator is quite variable. Therefore, authors should bring in enough arguments to link significantly their results with clinical.

Reviewer 2 Report

This is a very interesting and well executed study that will be of use to both clinicians and researchers in anatomical sciences. There are some shortcomings here and there that I believe the authors should address before this paper can be published. My key concern is the lack of discussion around natural human variation in the introduction. The authors should add a few sentences about how basic anatomical measures can vary between and within populations which could indicate pathologies but are not. 

My second suggestion is that the authors do not confuse gender with sex, unless they explicitly collected gender information from the patients (as in, gender identity). Otherwise, the authors are clearly referring to biological sex here.

My third suggestion is that parts of the Results section be moved either to the Introduction or the Discussion where the comparisons to other published data are made. The Results section should just present results.

Finally, there are just some minor presentation and writing mechanics issues here and there I encourage the authors to attend to.

I've attached the manuscript with my annotations in comments inserted into the PDF.

Thank you for the opportunity to review this manuscript. 

Reviewer 3 Report

Line 68-69: The conclusion of the sentence does not make any sence

Line 238-249: This paragraph belongs to the discussion, not in the results

Line 316-320: The differences in CHD values between the Western population and the present study are misleadingly explained. The first sentence describes no significant differences, the second that the distance was smaller. Here, the explanation should be improved again.

Round 2

Reviewer 1 Report

The authors did a good job and addressed all my comments.